# Dataset Inference for Data Provenance and Privacy Auditing in Tabular Foundation Models

**Dariush Wahdany** [* 1]  **Jesse C. Cresswell** [* 2]  **Naiqing Guan** [2]  **Atiyeh Ashari Ghomi** [2]
**Franziska Boenisch** [1]  **Adam Dziedzic** [1]

## Abstract

Tabular foundation models (TFMs) are increasingly deployed through black-box APIs and trained on real-world tabular datasets. While private, proprietary, or otherwise unauthorized datasets may be incorporated into pre-training corpora, there are currently no dedicated methods for determining whether a given tabular dataset was used to train a TFM. As a solution, we introduce the first dataset inference method for TFMs, aiming to infer whether a suspect dataset was part of a model's pre-training data. We systematically analyze a broad collection of candidate signals that can be observed from a black-box TFM via input manipulation and find that we can reliably infer dataset membership for several state-of-the-art TFMs trained on real tabular data, achieving up to 0.997 ROCAUC. We then study factors that influence dataset identification, including pre-training data composition, model capacity, and the use of real vs. synthetic data. Our results show that our dataset inference method is a practical auditing tool for detecting privacy leakage and the use of proprietary datasets in TFMs.

## 1. Introduction

Foundation models are a powerful paradigm for creating versatile models on various data modalities. The prototype, large language models (LLMs), use transformer architectures (Vaswani et al., 2017) to process vast amounts of pre-training data. In contrast to the rapid progress of this paradigm in the language domain, performance on predictive tabular data tasks has been dominated by gradient-boosted tree-based methods for most of the previous decade

(Chen & Guestrin, 2016; Ke et al., 2017; Prokhorenkova et al., 2018). Only with the rise of prior-data fitted networks (Müller et al., 2022; Hollmann et al., 2023; 2025), a transformer-based architecture which exhibits scaling properties similar to LLMs (Kaplan et al., 2020; Ma et al., 2025b), have tabular foundation models (TFMs) begun to convincingly outperform tree-based models (Qu et al., 2025; 2026; Erickson et al., 2025). Whereas traditional predictive models are fitted to a training dataset for inference on samples from the same distribution, TFMs are pre-trained on a wide variety of datasets. Once pre-trained, TFMs take a training dataset as context and directly predict on unlabeled test examples via in-context learning (Brown et al., 2020), without requiring updates to model weights. This ease of use, combined with their strong predictive power, has rapidly made TFMs a go-to tool for practitioners.

The first TFMs relied on synthetic data generated from a prior distribution over tabular datasets (Hollmann et al., 2023), which set the trend for many subsequent TFMs (Hollmann et al., 2025; Qu et al., 2025; Grinsztajn et al., 2025; Zhang et al., 2025c; Bouadi et al., 2025; Zhang et al., 2025b; Qu et al., 2026). However, scaling laws for TFMs (Ma et al., 2025b) indicate that real-world datasets can lead to better scaling exponents. Hence, recently, there has been increased effort from the research community in this direction (Vanschoren et al., 2014; Eggert et al., 2023; Spinaci et al., 2025).

Unfortunately, the drive to curate ever larger datasets for foundation model consumption has created controversy over ownership and licensing for data scraped from the internet (The New York Times, 2023; Carlini et al., 2023; Freeman et al., 2024; Ross et al., 2025). In the tabular space, further pressure has been added due to the rapid commercialization of TFMs hosted as black-box APIs (Prior Labs, 2025; Fundamental, 2025; Kumo AI, 2025; Neuralk-AI, 2026). Creators of black-box TFMs may have an incentive to train on as much real-world tabular data as possible, without diligently obtaining permission from the respective data owners or subjects. This lack of training data transparency creates an auditing problem for both model developers and external data owners. Organizations training or fine-tuning TFMs on sensitive internal data need to assess the risk of data leakage before releasing the model. Conversely, data owners will

---

[*]Equal contribution  [1]CISPA Helmholtz Center for Information Security  [2]Layer 6 AI, Toronto, Canada. Correspondence to: Dariush Wahdany <dariush.wahdany@cispa.de>, Jesse C. Cresswell <jesse@layer6.ai>.

*Proceedings of the $2^{nd}$ ICML Workshop on Foundation Models for Structured Data*, Seoul, South Korea. 2026. Copyright 2026 by the author(s).

want to verify whether their datasets were incorporated into a TFM without authorization. However, training-data identification for TFMs is challenging: the models have diverse architectures, attention mechanisms, and may be trained on datasets that are repeatedly augmented and redefined into different prediction tasks, in the manner of self-supervised learning (Chen et al., 2020; Balestriero et al., 2023; Ma et al., 2025b). These training procedures can substantially weaken direct pre-training evidence.

For these reasons, we investigate the feasibility of *dataset inference* (DI) for TFMs. Similar to membership inference (MI) (Shokri et al., 2017; Carlini et al., 2022), which aims to determine whether an individual data point was part of a model's training set, DI asks whether an entire dataset was included in the training corpus of a model (Maini et al., 2021). TFMs differ from other model types because their predictions depend jointly on model parameters, provided context dataset, and unlabeled query points. This creates new challenges and opportunities for DI. In this work, we design the first dataset inference method for TFMs. Our main contributions are as follows:

1. We formulate dataset inference (DI) for TFMs and design TFM-specific dataset membership signals.
2. We show that individual signals are weak, but multi-signal DI accurately identifies training datasets.
3. We find that real data is more detectable than prior-synthetic data.

## 2. Background

**Tabular Foundation Models**  Tabular foundation models (TFMs) differ widely in architecture, data processing, and training, but share an in-context learning (ICL) prediction interface. A TFM $f_\theta$ takes a labeled context dataset $\mathcal{D}_{\text{ctx}} = \{X_{\text{ctx}}, y_{\text{ctx}}\}$ and unlabeled query points $\mathcal{D}_{\text{qy}} = \{X_{\text{qy}}\}$ from the same distribution, and predicts $y_{\text{qy}}$ without updating its weights (Figure 1). TFMs may support classification, regression, or both. Early TFMs were trained mainly on synthetic data from priors (Hollmann et al., 2023), but recent work shows improved scaling from real tabular data (Ma et al., 2025b; Spinaci et al., 2025; Garg et al., 2025), and organizations increasingly train or adapt TFMs on private domain data (Breejen et al., 2024). This makes pre-training data leakage and provenance an increasingly relevant concern. Prior work has studied differentially private tabular prompting (Carey et al., 2024) and privacy risks in tabular generation (Byun et al., 2025); neither addresses pre-training dataset identification for predictive TFMs.

**Membership and Dataset Inference.**  Membership inference asks whether an individual data point was used to train a model (Shokri et al., 2017; Carlini et al., 2022), but becomes less reliable for large foundation models (Duan et al., 2024; Das et al., 2025; Maini et al., 2024). Dataset infer-

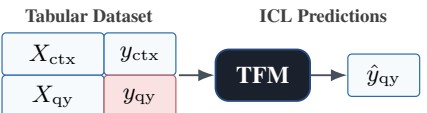

*Figure 1.* **Predictions with TFMs.**

ence (DI) instead asks whether an entire dataset was used during training (Maini et al., 2021). DI has been studied for self-supervised vision models (Dziedzic et al., 2022), LLMs (Maini et al., 2024; Rossi et al., 2026; Zhao et al., 2025), and generative image models (Dubiński et al., 2025; Kowalczuk et al., 2025). These methods typically extract per-example membership signals and aggregate them across a suspect dataset. TFMs differ fundamentally: they are predictive models trained on collections of datasets, and their outputs depend jointly on model parameters, context data, and query points. Thus, the relevant membership unit is a whole dataset, and existing DI methods do not directly apply. We design the first TFM-specialized DI method.

## 3. Problem Setup

Our work targets the question: *Was a given dataset used to train a TFM?* This question is relevant for both external and internal audits. These setups are conceptually similar, but differ in motivation and auditor knowledge.

**External Audits – Data Provenance.** External parties who own tabular datasets and suspect that these datasets were used to train a TFM $f_\theta$ without consent would like to verify this suspicion. They only have black-box access to the TFM, deployed behind an API. They hold the suspect dataset $\mathcal{D}_{\text{sus}}$ and some additional in-distribution datasets known to be non-members $\{\mathcal{D}_{\text{known}}\}$, such as unreleased private datasets or datasets randomly generated from a prior, as commonly used to train TFMs (Hollmann et al., 2023).

**Internal Audits – Privacy Leakage.** Organizations may train or fine-tune TFMs on proprietary datasets from sensitive domains such as finance, e-commerce, and healthcare. Before public deployment, they may need to audit whether sensitive information could leak through a hosted black-box API. Compared to an external auditor, an internal auditor has additional knowledge about how the TFM $f_\theta$ was trained. We assume they still access only the black-box API, but know the pre-training dataset members and non-member datasets, $\{\mathcal{D}_{\text{known}}\}$.

We consider DI for both external and internal audits, with the external setting being more challenging because it assumes less knowledge about the TFM's pre-training datasets.

## 4. Dataset Inference for TFMs

Our DI for TFMs asks whether a labeled *suspect* dataset $\mathcal{D}_{\text{sus}}$ was used to train a black-box TFM $f_\theta$. Following prior DI work (Maini et al., 2021; Dziedzic et al., 2022; Maini et al., 2024), the auditor also has additional *known* datasets

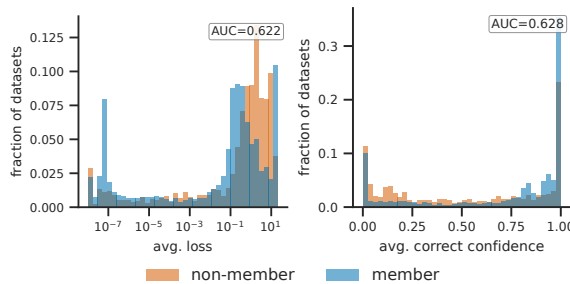

*Figure 2.* **Classical inference signals are weak.** Per-dataset average loss and correct-class confidence only weakly separate member and non-member datasets of SAP-RPT-OSS ($n$=19,888).

$\{\mathcal{D}_{\text{known}}\}$: all non-members for external audits, and a mix of members and non-members for internal audits. The auditor controls the TFM inputs, namely query points $X_{\text{qy}}^i$ and a context dataset $\mathcal{D}_{\text{ctx}}$, and observes the output distribution $p(y_{\text{qy}}^i \mid X_{\text{qy}}^i; \mathcal{D}_{\text{ctx}})$. Using $f_\theta$, the auditor maps each dataset $\mathcal{D}$ to a fingerprint vector $\bar{S}(\mathcal{D}) \in \mathbb{R}^d$, both for $\mathcal{D}_{\text{sus}}$ and for $\{\mathcal{D}_{\text{known}}\}$. Unlike DI in language or vision models, which typically extracts per-sample signals and aggregates them over a dataset, TFM pre-training is organized at the dataset level; hence, we treat $\mathcal{D}$ itself as the unit of membership. Given $\bar{S}(\mathcal{D}_{\text{sus}})$ and $\{\bar{S}(\mathcal{D}_{\text{known}})\}$, the auditor then decides whether $\mathcal{D}_{\text{sus}}$ was used for pre-training.

### 4.1. Dataset Fingerprints

Constructing $\bar{S}(\mathcal{D})$ is the key challenge. Standard membership signals such as loss or confidence (Shokri et al., 2017; Carlini et al., 2022) are only weak predictors for TFMs (Figure 2), likely because TFMs see many datasets under subsampling, shuffling, and task redefinitions during pre-training. We therefore exploit the auditor's main lever: full control over the ICL inputs $X_{\text{qy}}^i$ and $\mathcal{D}_{\text{ctx}}$.

Each coordinate of $\bar{S}(\mathcal{D})$ is a *(manipulation, signal, reduction)* triple. A manipulation perturbs the TFM inputs, a signal reads a scalar from the TFM output, and a reduction collapses the signal values across manipulation parameters into one feature. We use 14 output signals, 19 input manipulations, and 11 reductions; full details and examples are given in Appendix C. In short, signals include losses, confidences, and distributional quantities; manipulations include changing the context/query split, corrupting labels, shuffling rows or columns, and copying query rows into context; reductions summarize how the signal changes across parameter settings, e.g., by slope, AUC, mean, or stability.

### 4.2. External Auditing

For external auditing, $\{\mathcal{D}_{\text{known}}\}$ only contains non-members. Since no known members are available, we frame DI as anomaly detection calibrated on non-members (Figure 3). We fit an Isolation Forest (Liu et al., 2008; 2012) on $\{\bar{S}(\mathcal{D}_{\text{known}})\}$ and score $\bar{S}(\mathcal{D}_{\text{sus}})$ as an anomaly; datasets outside the normality region are flagged as likely members.

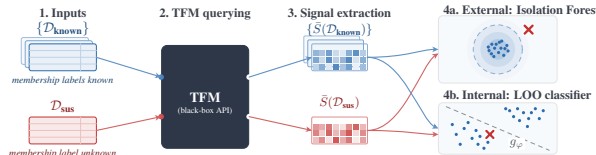

*Figure 3.* **Overview of our DI procedure.** The auditor queries the black-box TFM with known datasets $\{\mathcal{D}_{\text{known}}\}$ and a suspect dataset $\mathcal{D}_{\text{sus}}$, extracts fingerprints, and performs either anomaly detection or supervised membership classification.

For evaluation, we score held-out suspect datasets balanced between members and non-members. Given Isolation Forest scores $S_{\text{known}} = \{s_i\}_{i=1}^n$ on $\{\mathcal{D}_{\text{known}}\}$, we convert a suspect score $s$ into the conformal $p$-value

$$p(s) = \left(1 + \sum_{i=1}^n \mathbb{1}[s_i \geq s]\right)/(n+1).$$

To separate true membership leakage from distributional differences between members and non-members, we also report a blind baseline (Das et al., 2025): the same pipeline is run on a "blind" TFM with disjoint pre-training data, keeping all dataset labels unchanged.

### 4.3. Internal Auditing

For internal auditing, $\{\mathcal{D}_{\text{known}}\}$ contains both members and non-members. Given fingerprints $\{\bar{S}(\mathcal{D}_{\text{known}})\}$ and labels $y_i \in \{0, 1\}$, we train a binary classifier $g_\varphi$ and apply it to $\bar{S}(\mathcal{D}_{\text{sus}})$. We evaluate with leave-one-out cross-validation: each dataset is treated once as $\mathcal{D}_{\text{sus}}$, while all others form $\{\mathcal{D}_{\text{known}}\}$ for training. We report ROCAUC over the left-out predictions. As in the external setting, we compare to a blind TFM baseline; excess accuracy over this baseline indicates genuine dataset membership leakage.

## 5. Experimental Evaluation

### 5.1. Experimental Setup

We evaluate DI on three target TFMs: TabDPT (Ma et al., 2025b), SAP-RPT-1-OSS (Spinaci et al., 2025), and nanoTabPFN (Pfefferle et al., 2025). For TabDPT and SAP-RPT-1-OSS, we use released checkpoints; for nanoTabPFN, we train with default settings. Each model was trained on a fixed corpus, which defines member datasets; all other datasets are non-members. TabDPT and SAP-RPT-1-OSS were trained on real tabular data, whereas nanoTabPFN was trained on fixed prior-synthetic datasets. Full dataset details are in Appendix D.1. As a blind baseline (Das et al., 2025), we use TabPFN-2.5 (Grinsztajn et al., 2025), which shares no fixed members with the target TFMs. For internal audits, we use logistic regression, XGBoost (Chen & Guestrin, 2016), and TabPFN-2.5 as membership classifiers $g_\varphi$; details are in Appendix D.2.

### 5.2. Dataset Inference Performance

**External auditing.** Table 1 reports ROCAUCs for the Isolation Forest calibrated only on known non-members.

*Table 1.* **External Audit.** ROCAUC of DI on suspect datasets using Isolation Forest. Excess over the blind baseline quantifies membership leakage beyond distributional differences.

| Target TFM | Target | Blind | Excess |
|---|---|---|---|
| nanoTabPFN | $0.499 \pm 0.001$ | $0.501 \pm 0.001$ | $-0.002$ |
| TabDPT | $0.612 \pm 0.041$ | $0.576 \pm 0.029$ | $+0.036$ |
| SAP-RPT-1-OSS | $0.851 \pm 0.028$ | $0.533 \pm 0.042$ | $+0.318$ |

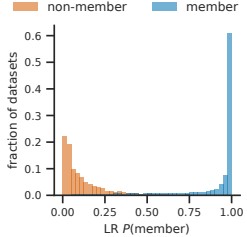

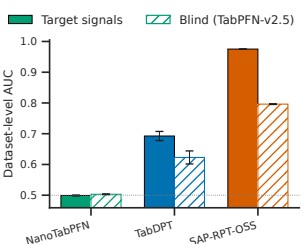

*Figure 4.* Multiple signals separate SAP-RPT-1-OSS members from non-members, unlike single signals (Figure 2).

*Figure 5.* **Internal Audit.** DI success ranges from no power on nanoTabPFN to near-perfect AUC on SAP-RPT-1-OSS.

SAP-RPT-1-OSS shows a strong membership signal, with AUC 0.851 and an excess of $+0.318$ over the blind baseline. TabDPT leaks only weakly, while nanoTabPFN shows no detectable leakage. Empirical $p$-value distributions for SAP-RPT-1-OSS are shown in Appendix D.3.

**Internal auditing.** Figure 4 shows that combining TFM-specific signals yields clear member/non-member separation on SAP-RPT-1-OSS, whereas individual loss or confidence signals were weak (Figure 2). Figure 5 reports internal-audit AUCs using TabPFN-2.5 as $g_\varphi$. Again, nanoTabPFN is indistinguishable from random, TabDPT leaks weakly, and SAP-RPT-1-OSS leaks strongly. Blind baselines can achieve non-trivial AUC, confirming that distributional differences matter; therefore, the excess over the blind baseline is the relevant measure of true membership leakage. Full results for all classifiers are in Appendix D.4.

Feature attribution via Shapley values (Shapley, 1953) shows that many manipulations contribute, but the most revealing signal on SAP-RPT-1-OSS is the chi-squared statistic, suggesting that correct output marginals are highly informative (Appendix D.6).

### 5.3. What Drives Inference Success?

NanoTabPFN differs from TabDPT and SAP-RPT-1-OSS in several ways: corpus size, model size, and the use of prior-synthetic training data. We ablate these three factors to uncover why nanoTabPFN does not leak under default training, while TabDPT and SAP-RPT-1-OSS do.

**Number of Datasets** Retraining nanoTabPFN on a smaller corpus only shows significant DI success at the extreme. When the default corpus with 50,000 datasets is reduced to 100, AUC reaches 0.72 (Table 2), but at this level TFM quality is also degraded.

*Table 2.* **Dataset-volume ablation.** Each row shows nanoTabPFN trained on a differently sized corpus of synthetic prior tables. DI is run on a set of suspects balanced between members and non-members. Top-1 is the best single-signal AUC; multivariate is logistic regression on the top 50 signals.

| $|\mathcal{D}_{\text{train}}|$ | top-1 AUC | multivariate AUC |
|---|---|---|
| 100 | 0.632 | $0.720 \pm 0.061$ |
| 500 | 0.570 | $0.563 \pm 0.037$ |
| 1,000 | 0.536 | $0.541 \pm 0.037$ |
| 5,000 | 0.531 | $0.523 \pm 0.009$ |
| 10,000 | 0.516 | $0.514 \pm 0.005$ |
| 50,000 | 0.505 | $0.501 \pm 0.003$ |

*Table 3.* **NanoTabPFN real/synthetic ablation.** Deafult nano-TabPFN trained on real data from T4, and on a Gaussian-copula synthetic copy of T4.

| Training corpus | top-1 AUC | multivariate AUC |
|---|---|---|
| Real | 0.811 | $0.923 \pm 0.006$ |
| Gaussian copula | 0.871 | $0.897 \pm 0.008$ |

**Model Size** We increase nanoTabPFN's capacity by adding more layers, increasing embedding dimensionality, adding attention heads, and increasing the MLP hidden dimension. The largest hyperparameter setting matches SAP-RPT-1-OSS. Overall, we vary model size between 3.72 M and 115.8 M parameters, but DI remains unsuccessful, see Table 11 in Appendix D.7.

**Real vs. synthetic** Replacing prior-synthetic data with real datasets from T4 (Gardner et al., 2024) increases DI AUC to 0.923, and real-derived Gaussian-copula synthetic data remains highly detectable (Table 3). We conclude that pre-training data composition is the main driver for membership leakage: real and real-derived datasets leave strong dataset-level traces, while diverse prior-synthetic data largely does not. Full ablations are in Appendix D.7.

## 6. Conclusion

In this work, we introduced dataset inference for tabular foundation models as a black-box auditing method for determining whether a suspect dataset was part of a model's pre-training corpus. By exploiting the in-context prediction interface of TFMs, we build dataset-level fingerprints from diverse input manipulations and output signals. While individual classical membership signals are weak, their combination can reliably reveal dataset membership for TFMs trained on real tabular data. Our ablations show that leakage is driven less by model capacity than by pre-training data composition: real and real-derived synthetic datasets leave strong dataset-level traces, whereas diverse prior-synthetic data is largely resistant. These results suggest that dataset inference can serve as a practical tool for provenance and privacy auditing of deployed TFM APIs.

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

## A. Limitations and Ethical Considerations

Our work has several limitations and dual-use implications. First, successful dataset inference requires aggregating many weak features: individual signals such as loss or confidence provide only limited evidence of membership. Second, our method is mainly relevant for TFMs trained on fixed corpora of real or real-derived tabular datasets; many current TFMs are trained on non-fixed synthetic data, where there may be no meaningful finite training corpus to audit, and our experiments show little to no inference success on fixed synthetic data except in extreme low-diversity regimes. Third, our approach is model-agnostic and does not exploit architecture- or training-specific details, so stronger attacks may be possible when more information about the audited TFM is available. Finally, while we frame dataset inference as a tool for provenance and privacy auditing, similar techniques could be misused to probe whether sensitive datasets contributed to a deployed API. We do not condone such use, but believe that studying these risks is necessary to help data owners and model providers assess leakage, improve transparency, and develop appropriate privacy safeguards.

## B. Extended Background on Tabular Foundation Models

In Section 2 we introduced the commonalities between existing TFMs, namely their ICL prediction mechanism. We also discussed the use of real and synthetic data for pre-training. Here, we give more detail on the diverse architectural details, data treatments, and training methods used for TFMs currently.

Internally, TFMs use an attention mechanism (Vaswani et al., 2017), where predictions on $X_{\text{qy}}^i$ can attend to context datapoints, but not to other query datapoints. The actual implementation varies greatly, with some choices being to attend across rows (datapoints indexed by $i$) (Hollmann et al., 2023; Ma et al., 2025b), across columns (elements of $X^i$), or some hybrid combination (Hollmann et al., 2025; Qu et al., 2025; Spinaci et al., 2025; Bouadi et al., 2025). These choices are further complicated due to the ways data can be encoded. Since tabular data is heterogeneous, where columns of $X$ could be floats, ints, dates, or categorical, among others, TFMs require an encoder as a first step to create a uniform vector representation. Choosing what and how to encode is a major design decision, but the most common choice is to tokenize entire rows (Hollmann et al., 2023; Ma et al., 2025b), as opposed to individual cells.

To enhance the variety of training tasks, the prediction target $y$ can be randomly chosen out of all columns in the dataset, a form of self-supervised learning (Ma et al., 2025b;a; Sui et al., 2024). This style of task creation has been successfully extended to foundation models for causal inference which also act on tabular data (Balazadeh et al., 2025; Stith et al., 2026).

We also note a separate line of research focused on tabular prediction using LLMs. Due to the general capabilities of LLMs to reason and perform ICL, efforts have been made to adapt them for tabular prediction tasks through ICL (Jaitly et al., 2023; Wen et al., 2024b), or by post-training LLMs on tabular prediction tasks (Wang et al., 2023; Gardner et al., 2024; Su et al., 2024; Zhang et al., 2025a; Wen et al., 2025; Wang et al., 2026). Because LLMs are pre-trained on vast quantities of non-tabular data and fundamentally process text tokens, we consider this setting to be distinct, and instead focus solely on TFMs.

*Table 4.* Output **signals** for dataset inference on TFMs. **C** - classification tasks; **R** - Regression tasks. $\hat{y}^i$ denotes the point prediction $\arg\max_y p(y)$ for query row $i$.

| Signal Name | Type | Formula | Description |
|---|---|---|---|
| **S1:** Correct class confidence | C | $p(y) := p(y \mid X_{qy}^i; \mathcal{D}_{ctx})$ | Likelihood assigned to the correct class |
| **S2:** Max confidence | C | $\max_y p(y)$ | Highest likelihood assigned to any label |
| **S3:** Prediction margin | C | $p(y_{(1)}) - p(y_{(2)})$ | Difference of highest two class likelihoods |
| **S4:** Accuracy | C | $\mathbb{1}(\hat{y}^i = y_{qy}^i)$ | Correctness of predicted class |
| **S5:** Cross entropy | C | $\sum_{k=1}^{K} y_{qy}^i \log p_k(y)$ | Cross entropy loss |
| **S6:** Absolute error | R | $\lvert \hat{y}^i - y_{qy}^i \rvert$ | Absolute error loss |
| **S7:** Mean squared error | R | $(\hat{y}^i - y_{qy}^i)^2$ | Mean squared error loss |
| **S8:** Squared Residuals | R | $1 - \sum_i (y_i - \hat{y}_i)^2 / \sum_i (y_i - \bar{y})^2$ | Coefficient of Determination |
| **S9:** Prediction Spread | R | $\sigma(\hat{y})/\sigma(y)$ | Relative standard deviation |
| **S10:** Chi-Squared$^\dagger$ | CR | $\sum_{k=1}^{K} (p(y)_k - q(y)_k)^2 / q(y)_k$ | Comparison of class frequencies to reference $q(y)$ |
| **S11:** Entropy | CR | $-\sum_y p(y) \log p(y)$ | Shannon entropy of class probabilities |
| **S12:** KL Divergence | CR | $\sum_y p(y) \log \frac{p(y)}{q(y)}$ | KL Divergence to a reference distribution $q(y)$ |
| **S13:** Reverse KL | CR | $\sum_y q(y) \log \frac{q(y)}{p(y)}$ | KL Divergence from a reference distribution $q(y)$ |
| **S14:** Wasserstein Distance | CR | $W_1(\hat{y}, y)$ | Wasserstein-1 to reference output $y$ |

$\dagger$: Computed in 30 bins for **R**.

## C. Dataset Inference Details

Here we give the details omitted from the main method. For each dataset $\mathcal{D}$, we construct a fingerprint $\bar{S}(\mathcal{D})$ by querying the TFM under multiple structured input manipulations and recording output-derived signals. Each fingerprint coordinate corresponds to one valid *(manipulation, signal, reduction)* triple.

**Signals.** Signals are scalar functions of the TFM output distribution $p(y_{qy}^i \mid X_{qy}^i; \mathcal{D}_{ctx})$, aggregated over query points. We include classical membership-inference quantities such as correct-class confidence, maximum confidence, prediction margin, accuracy, and loss, as well as distributional quantities such as entropy, KL divergence, reverse KL, chi-squared distance, and Wasserstein distance. Since the auditor does not know the TFM architecture or training objective, we use a broad collection of signals rather than assuming the exact loss used during pre-training. The full list is given in Table 4.

**Manipulations.** Manipulations perturb the TFM inputs $X_{qy}^i$ and $\mathcal{D}_{ctx}$ in ways that exploit the ICL prediction interface. The intuition is that if $\mathcal{D}$ was used during pre-training, $f_\theta$ may have internalized some parametric information about the relation between $X$ and $y$, which can change its behavior under unusual contexts or perturbations. Our manipulations include varying the context/query split, subsampling rows or columns, shuffling rows or features, adding feature noise, changing the prediction task, permuting or corrupting labels, duplicating rows, copying query rows into context, and nearest-neighbor retrieval. The full list is given in Table 5.

Two examples illustrate the design. **Query only** splits $\mathcal{D}$ but discards $\mathcal{D}_{ctx}$, querying the model only with $X_{qy}$. If $\mathcal{D}$ is a member, the TFM may still predict some labels correctly from parametric knowledge; for non-members, predictions should be closer to random. **Brainwash** copies some query rows into the context with corrupted labels. For non-members, the model should mainly follow the corrupted context, while for members, parametric knowledge may partially overcome it. This is inspired by the Brainwash attack (Wen et al., 2024a).

*Table 5.* Input **manipulations** for dataset inference on TFMs and their parameters. **C** - classification tasks; **R** - Regression tasks

| Manipulation Name | Type | Description | Parameters |
|---|---|---|---|
| **M1:** No manipulation | **CR** | Only split rows of $\mathcal{D}_{\text{sus}}$ into disjoint sets $(X_{\text{ctx}}, y_{\text{ctx}})$ and $(X_{\text{qy}}, y_{\text{qy}})$ | Splitting fraction (applies to all below) |
| **M2:** Query = Context | **CR** | Repeat the context inputs $X_{\text{ctx}}$ as $X_{\text{qy}}$ | - |
| **M3:** Query only | **CR** | Provide only $X_{\text{qy}}$, with $\mathcal{D}_{\text{ctx}} = \varnothing$ | - |
| **M4:** Row sampling | **CR** | Subsample the rows of $\mathcal{D}_{\text{ctx}}$ | Fraction of rows sampled |
| **M5:** Row shuffling | **CR** | Randomly permute the rows of $\mathcal{D}_{\text{ctx}}$ | Permutation |
| **M6:** Feature sampling | **CR** | Subsample the columns of $X_{\text{ctx}}$ | Fraction of columns sampled |
| **M7:** Feature shuffling | **CR** | Randomly permute the columns of $X_{\text{ctx}}$ | Permutation |
| **M8:** Feature noising | **CR** | Randomly perturb the columns of $\tilde{X}_{\text{ctx}} = X_{\text{ctx}} + \varepsilon$ | Noise type (e.g. Gaussian); Noise level |
| **M9:** Feature transform | **CR** | Transform a column ($\mathcal{D}_{\text{ctx}}$ and $\mathcal{D}_{\text{qy}}$) with a fixed function | Column; Function (e.g. log) |
| **M10:** Task swap | **CR** | Redefine the label $y$ as one of the columns of $X$ | Column |
| **M11:** Class permutation | **C** | Re-encode classes as $y' = \pi(y)$ with a permutation $\pi$ | Permutation |
| **M12:** Label corruption | **CR** | Randomly select context rows and change $y_{\text{ctx}}$ to an incorrect label | Fraction corrupted |
| **M13:** Constant labels | **CR** | Set all context labels to a fixed value $y_{\text{ctx}} = a$ | Value |
| **M14:** Row duplication | **CR** | Randomly duplicate some rows of $\mathcal{D}_{\text{ctx}}$ | Fraction duplicated |
| **M15:** Truth serum (Tramèr et al., 2022) | **CR** | Randomly duplicate some rows of $\mathcal{D}_{\text{ctx}}$, but with corrupted $y_{\text{ctx}}$ | Fraction duplicated |
| **M16:** Query leakage | **CR** | Randomly copy some rows of $\mathcal{D}_{\text{qy}}$ into $\mathcal{D}_{\text{ctx}}$ | Fraction copied |
| **M17:** Brainwash (Wen et al., 2024a) | **CR** | Randomly copy some rows of $\mathcal{D}_{\text{qy}}$ into $\mathcal{D}_{\text{ctx}}$, but with corrupted $y_{\text{ctx}}$ | Fraction copied |
| **M18:** Nearest neighbours | **CR** | Compute NN within $\mathcal{D}$, and split them across $\mathcal{D}_{\text{ctx}}$ and $\mathcal{D}_{\text{qy}}$ | Number of neighbours |
| **M19:** Retrieval | **CR** | Retrieve NN of $X_{\text{qy}}$ from $\mathcal{D}$ as $X_{\text{ctx}}$ | Number of neighbours |

**Reductions.** Many manipulations have parameters, such as a corruption fraction, noise level, context size, or random permutation. For each manipulation–signal pair, varying these parameters yields a sequence of signal values, which we reduce to a scalar feature. For ordered parameters, we use curve summaries such as slope, area under the curve, gain, early value, and late gain. For unordered parameters, we use statistics such as mean, standard deviation, and range. For context-only manipulations, predictions for the same query row can be compared across parameter settings, allowing stability reductions such as consistency and rank correlation. The full list is given in Table 6.

As a concrete example, the feature `brainwash_loss_slope` applies the Brainwash manipulation at multiple corruption fractions, records the loss signal at each fraction, and stores the slope of the resulting curve as one coordinate of $\bar{S}(\mathcal{D})$.

*Table 6.* **Reductions** vary the parameters of a manipulation to create a sequence of signals that are then summarised into a scalar entry of $\bar{S}(\mathcal{D})$. The applicable family depends on the structure of the manipulation's parameters: *ordered* along an intensity axis, *unordered*, or based on prediction *stability*.

| Type | Reduction | Formula | Captures |
|---|---|---|---|
| Default | **R1:** No Reduction | Signal is used directly as feature | Predictive power of signal directly |
| Ordered | **R2:** Slope | Regression coef. on (log-spaced) intensity | Rate of signal change as manipulation intensity increases |
| | **R3:** AUC | Normalised trapezoidal integral | Average signal value across manipulation intensity range |
| | **R4:** Gain | Final value minus initial value | Total signal change between highest and lowest intensity |
| | **R5:** Early | Value at 25% intensity, normalised by max | Signal response at low-intensity manipulation |
| | **R6:** Late gain | Final value minus second-to-last | End-of-curve flattening or runaway signal |
| Unordered | **R7:** Mean | Mean over parameter range | Typical signal value |
| | **R8:** St. dev. | Sample st. deviation over parameter range | Signal spread |
| | **R9:** Range | Max minus min over parameter range | Signal range |
| Stability | **R10:** Consistency | Pairwise signal agreement | Cross-parameter signal stability |
| | **R11:** Rank corr | Mean pairwise Spearman correlation across per-row signals | Cross-parameter ranking stability |

## C.1. Additional Audit Details

**External auditing.** In the external setting, all known datasets $\{\mathcal{D}_{\text{known}}\}$ are assumed to be non-members. We fit an Isolation Forest (Liu et al., 2008; 2012) on their fingerprints $\{\bar{S}(\mathcal{D}_{\text{known}})\}$ to characterize typical non-member behavior. A suspect dataset $\mathcal{D}_{\text{sus}}$ is inferred as more likely to be a member when its fingerprint $\bar{S}(\mathcal{D}_{\text{sus}})$ receives a more anomalous score. This setup requires neither known member datasets nor shadow-model training (Shokri et al., 2017).

**Internal auditing.** In the internal setting, the auditor knows the ground-truth membership labels of the known datasets. We therefore train a binary classifier $g_\varphi$ on fingerprints $\{\bar{S}(\mathcal{D}_{\text{known}})\}$ with labels $y_i \in \{0, 1\}$. In our experiments, we evaluate this procedure with leave-one-out cross-validation, where each dataset is held out once as the suspect dataset and the classifier is trained on all remaining fingerprints. We report the ROCAUC over these held-out predictions.

**Blind baselines.** For both audit settings, we use blind baselines to control for distributional differences between member and non-member datasets. The blind baseline replaces the audited TFM with another TFM that does not share the relevant pre-training datasets, while keeping the same suspect datasets and membership labels. Any performance achieved by the blind baseline cannot be attributed to true membership leakage from the target TFM; therefore, the excess over the blind baseline is the relevant DI signal.

# D. Additional Experimental Details and Results

## D.1. Tabular Foundation Models and Datasets

Our DI experiments used three TFMs: TabDPT (Ma et al., 2025b) (Apache 2.0 license), SAP-RPT-1-OSS (Spinaci et al., 2025) (Apache 2.0 license), and nanoTabPFN (Pfefferle et al., 2025) (Apache 2.0 license via TFM-Playground (AutoML, 2026)). Each model was trained on a different fixed set of datasets, which defines its members.

**TabDPT** (Ma et al., 2025b) was trained on a collection of 123 datasets from OpenML (Vanschoren et al., 2014) (CC-BY license). For held-out non-members, we use CC18 (Bischl et al., 2021) and CTR23 (Fischer et al., 2023), also from OpenML, which TabDPT used as test datasets rather than pre-training data.

**SAP-RPT-1-OSS** (Spinaci et al., 2025) was trained on a subset of the T4 dataset (Gardner et al., 2024) (available under a custom license). This subset excluded datasets with fewer than 150 rows, leaving 2.18M tables available for pre-training. Since this number is excessively large for testing DI, we select tables in chunk-002 from T4 with more than 150 rows as members. Datasets with 150 or fewer rows are known non-members. Given this setup, DI would be trivial if row count were used as a predictive feature; hence, we exclude any feature related to row count or table size.

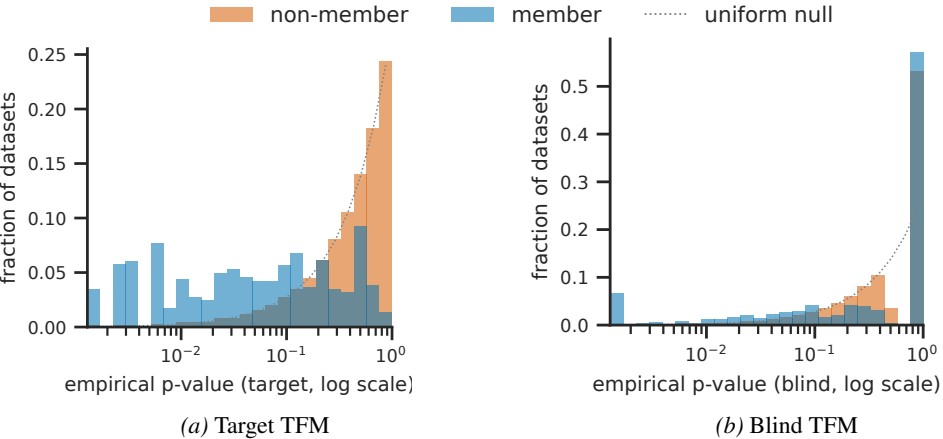

*(a) Target TFM*        *(b) Blind TFM*

*Figure 6.* **External audit DI.** Empirical $p$-values from the Isolation Forest fit on $\{\bar{S}(\mathcal{D}_{\text{known}})\}$ and evaluated on suspect datasets $\mathcal{D}_{\text{sus}}$. Members concentrate in the low-$p$ tail on the target model but not on the blind model.

**nanoTabPFN** (Pfefferle et al., 2025) and its pre-training datasets are available in the TFM-Playground repository (AutoML, 2026) (Apache 2.0 license). Using default settings, this model is trained on synthetic data from a prior. Unlike TFMs that generate new synthetic data for each batch (Grinsztajn et al., 2025), nanoTabPFN trains on a fixed set of synthetic tables that are recycled each epoch. The fixed set consists of 690k generated tables as members and another 690k as non-members.

Many other TFMs are not trained on a fixed set of datasets, but instead draw new synthetic datasets from a prior for each batch during pre-training. Such synthetic datasets generally do not come with privacy or provenance concerns, so we do not consider DI on these TFMs. However, this makes them suitable as blind baselines. In particular, we use the highly performant **TabPFN-2.5** (Grinsztajn et al., 2025), with model weights released under an Apache 2.0 license, as our blind baseline.

### D.2. Additional Experimental Details

For internal auditing, we evaluate three classifiers $g_\varphi$ on the dataset fingerprints: logistic regression, XGBoost (Chen & Guestrin, 2016), and TabPFN-2.5 (Grinsztajn et al., 2025). We use a TFM as one possible $g_\varphi$ because TFMs are strong tabular classifiers. This classifier is separate from the audited target TFM $f_\theta$ and from the blind baseline. When $g_\varphi$ is TabPFN-2.5, it is not trained by gradient descent; instead, it uses $\{\bar{S}(\mathcal{D}_{\text{known}})\}$ as context and $\bar{S}(\mathcal{D}_{\text{sus}})$ as query.

Unless otherwise stated, reported uncertainties are standard deviations over repeated trials.

### D.3. External Audit $p$-Values

Figure 6 shows empirical $p$-values produced by the external anomaly detection method on SAP-RPT-1-OSS. On the target TFM, many member datasets concentrate in the low-$p$ tail, while non-members closely follow the uniform reference. On the blind TFM, both member and non-member datasets cluster near $p \approx 1$, as expected when the model has seen neither $\mathcal{D}_{\text{sus}}$ nor $\mathcal{D}_{\text{known}}$ during pre-training.

### D.4. Internal Audit Results

Table 7 reports the full internal-audit ROCAUC results for all target TFMs and membership classifiers. The blind TFM row reports the same pipeline run against fingerprints from TabPFN-2.5. The excess over the blind baseline is the relevant DI signal, since it accounts for distributional differences between member and non-member datasets.

### D.5. Signal Separation on TabDPT

Figure 7 reproduces the single-signal separation analysis on TabDPT (`binary_all_tabdpt`, $n$=224). Single-signal AUCs are again close to chance (0.61 for both average loss and average correct confidence), while the multi-signal logistic regression reaches 0.70.

*Table 7.* **Internal-audit DI AUC by classifier.** LOOCV ROCAUC of dataset-membership prediction. Excess over the blind baseline quantifies the DI signal attributable to the target TFM.

| Target TFM $f_\theta$ ↓ | Classifier $g_\varphi$ → | XGB | LR | TabPFN-2.5 |
|---|---|---|---|---|
| nanoTabPFN | Target TFM | $0.497 \pm 0.002$ | $0.500 \pm 0.001$ | $0.503 \pm 0.002$ |
| | Blind TFM | $0.498 \pm 0.001$ | $0.503 \pm 0.001$ | $0.499 \pm 0.001$ |
| | Excess | $-0.001 \pm 0.002$ | $-0.003 \pm 0.001$ | $+0.005 \pm 0.002$ |
| TabDPT | Target TFM | $0.664 \pm 0.033$ | $0.608 \pm 0.019$ | $0.691 \pm 0.029$ |
| | Blind TFM | $0.603 \pm 0.029$ | $0.607 \pm 0.016$ | $0.615 \pm 0.019$ |
| | Excess | $+0.060 \pm 0.043$ | $+0.001 \pm 0.024$ | $+0.077 \pm 0.026$ |
| SAP-RPT-1-OSS | Target TFM | $0.992 \pm 0.000$ | $0.955 \pm 0.000$ | $0.997 \pm 0.000$ |
| | Blind TFM | $0.824 \pm 0.001$ | $0.792 \pm 0.001$ | $0.831 \pm 0.002$ |
| | Excess | $+0.168 \pm 0.001$ | $+0.163 \pm 0.001$ | $+0.166 \pm 0.002$ |

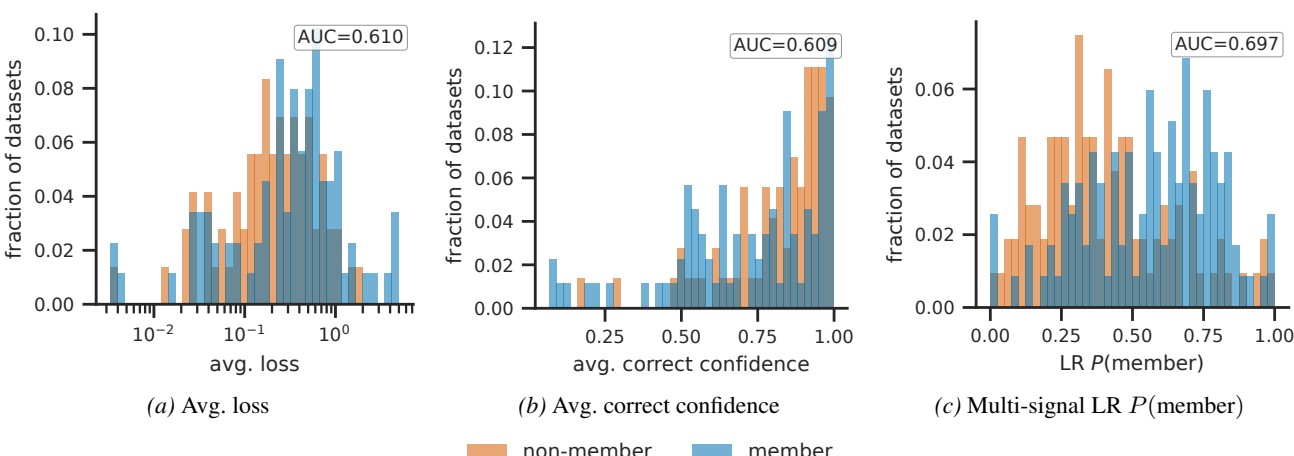

*(a)* Avg. loss     *(b)* Avg. correct confidence     *(c)* Multi-signal LR $P(\text{member})$

non-member     member

*Figure 7.* **Signal separation on TabDPT.** Per-dataset distributions of two common inference signals and of the multi-signal LR score for TabDPT ($n=224$).

### D.6. Fingerprint Feature Contributions

To identify which fingerprint features drive membership prediction, we compute Shapley values (Shapley, 1953) for the XGBoost classifier used in internal auditing. Table 8 shows the top features for SAP-RPT-1-OSS compared to the blind baseline. A range of manipulations and reductions contribute, emphasizing the value of controlling the TFM inputs. The chi-squared signal appears especially informative, indicating that correct label marginals or binned regression marginals are a strong dataset-level trace.

Table 9 reports the top Shapley-contributing features for TabDPT. Compared to SAP-RPT-1-OSS, the overall DI signal is weaker, but the most informative features still involve combinations of TFM-specific manipulations, losses, and distributional signals.

### D.7. Ablations

We use nanoTabPFN to isolate factors that drive DI success, since it is modular and lightweight to train. Table 10 reports compute. We vary model size, the number of synthetic pre-training datasets, and the source of pre-training data. Increasing model size alone does not produce leakage (Table 11); reducing synthetic dataset diversity only matters in the extreme low-data regime (Table 2); and replacing prior-synthetic data with real or real-derived data creates a strong DI signal (Table 3).

**Number of synthetic datasets.** We decrease the number of pre-training datasets while keeping the total number of tables seen during training fixed, thereby increasing each dataset's relative exposure. Table 2 shows that leakage appears only in the extreme low-diversity regime: with 100 synthetic member datasets, multivariate AUC increases to 0.720, but quickly returns close to chance as the number of datasets grows.

*Table 8.* **Fingerprint feature contributions on SAP-RPT-1-OSS.** Mean absolute Shapley value of the top 10 features from $\bar{S}(\mathcal{D})$ for SAP-RPT-1-OSS with the XGB classifier. Identifiers reference Tables 4 to 6.

| Fingerprint Feature | Manip. | Signal | Red. | SAP-RPT | Blind |
|---|---|---|---|---|---|
| Retrieval_ChiSquared_Mean | M19 | S10 | R7 | 0.6409 | 0.0009 |
| TaskSwap_ChiSquared_Mean | M10 | S10 | R7 | 0.4718 | 0.0156 |
| NN_ChiSquared_Default | M18 | S10 | R1 | 0.3287 | 0.0033 |
| RowSampling_ChiSquared_Gain | M4 | S10 | R4 | 0.3245 | 0.0821 |
| ClassPermutation_ChiSquared_Range | M11 | S10 | R9 | 0.3154 | 0.0652 |
| RowSampling_ChiSquared_Slope | M4 | S10 | R2 | 0.3104 | 0.0449 |
| ClassPermutation_ChiSquared_Mean | M11 | S10 | R7 | 0.2910 | 0.1389 |
| Retrieval_KLD_Mean | M19 | S12 | R7 | 0.2735 | 0.0039 |
| LabelCorruption_ChiSquared_Gain | M12 | S10 | R4 | 0.2480 | 0.0044 |
| Brainwash_ReverseKL_Early | M17 | S13 | R5 | 0.2283 | 0.0021 |

*Table 9.* **Per-signal Shapley contributions on TabDPT.** Mean absolute Shapley value of the top 10 features on TabDPT's pre-training dataset using TabDPT as the target model and TabPFN-2.5 as the blind model. Larger values indicate signals that contribute more to membership prediction.

| | Signal | TabDPT | Blind |
|---|---|---|---|
| $s_1$ | col_kl_divergence_auc | 0.4358 | 0.0000 |
| $s_2$ | splitsize_loss_early | 0.4125 | 0.0000 |
| $s_3$ | ctx_kl_divergence_early | 0.4018 | 0.0000 |
| $s_4$ | splitsize_correct_confidence_slope | 0.3576 | 0.0000 |
| $s_5$ | noise_chi_squared_auc | 0.1962 | 0.0225 |
| $s_6$ | splitsize_loss_slope | 0.1838 | 0.0000 |
| $s_7$ | col_wasserstein_late_gain | 0.1810 | 0.0000 |
| $s_8$ | bwash_loss_late_gain | 0.1795 | 0.0000 |
| $s_9$ | bwash_chi_squared_auc | 0.1766 | 0.0102 |
| $s_{10}$ | qleak_kl2clean_early | 0.1723 | 0.0133 |

**Model size.** We increase nanoTabPFN's capacity by adding more layers, increasing embedding dimensionality, adding attention heads, and increasing the MLP hidden dimension. The largest configuration is approximately $32\times$ larger than the default model. As shown in Table 11, increased model size alone does not lead to successful DI.

**Real vs. synthetic data.** We replace synthetic prior datasets with real data from T4 (Gardner et al., 2024). We also train on real-derived synthetic data generated with a Gaussian Copula synthesizer. The synthesizer fits an empirical CDF per column, maps columns to a standard normal scale, fits a multivariate Gaussian, samples from it, and maps samples back to the original column space. This preserves marginals and pairwise correlations. As Table 3 shows, real data leads to strong DI success, and real-derived synthetic data remains highly detectable. This suggests that dataset-level statistical structure, not only individual rows, can drive membership leakage.

*Table 10.* **NanoTabPFN training cost.** Single-GPU (NVIDIA H100) training time for the default checkpoint and all ablation variants used in the paper. Architecture is given as L / E / H / M (layers, embedding dim, attention heads, MLP hidden dim). All variants are trained for 80 epochs of 25 steps each at batch size 50 with AdamW ($\eta = 10^{-4}$).

| Variant | L/E/H/M | Prior tables | Time/epoch | Compute |
|---|---|---|---|---|
| `default` | 6/192/6/768 | 50 000 | 0.55 s | 44 s |
| `data25k` | 6/192/6/768 | 25 000 | 0.54 s | 43 s |
| `data10k` | 6/192/6/768 | 10 000 | 0.52 s | 42 s |
| `data5k` | 6/192/6/768 | 5 000 | 0.52 s | 42 s |
| `arch_medium` | 8/384/8/1536 | 50 000 | 1.34 s | 1.8 min |
| `arch_large` | 12/768/12/3072 | 50 000 | 5.35 s | 7.1 min |

*Table 11.* **NanoTabPFN architecture ablation.** All variants are trained on the same 50 000 synthetic prior tables; only the parameter count differs. "L/E/H/M" denotes layers, embedding dim, attention heads, and feed-forward dim. DI evaluation is run on the same held-out 50 000 member / 50 000 non-member synthetic split.

| Variant | L/E/H/M | params | eval $n$ | top-1 AUC | multivariate AUC |
|---|---|---|---|---|---|
| Default (small) | 6/192/6/768 | 2.7M | 100,000 | 0.505 | $0.501 \pm 0.003$ |
| Medium | 8/384/8/1536 | 14.2M ($\times 5$) | 100,000 | 0.506 | $0.505 \pm 0.003$ |
| Large | 12/768/12/3072 | 84.9M ($\times 32$) | 100,000 | 0.505 | $0.503 \pm 0.004$ |

