# OpenReview forum: "Dataset Inference for Data Provenance and Privacy Auditing in Tabular Foundation Models"
_ICML.cc/2026/Workshop/FMSD — FMSD @ ICML 2026 SpotlightOral_

### Official Review · Reviewer_q56M · 2026-05-16
**Strong Privacy Audit Study with Slight Practical Caveats**

**Rating:** 7
**Confidence:** 4

**Review:**

## Summary

This paper introduces dataset inference for tabular foundation models: given black-box access to a TFM and a suspect dataset, the goal is to infer whether that dataset was part of the model’s pretraining corpus. The method builds dataset-level fingerprints by querying the TFM under many input manipulations and output signals, then uses anomaly detection for external audits or supervised classification for internal audits. The paper evaluates the method on TabDPT, SAP-RPT-1-OSS, and nanoTabPFN, finding strong dataset-inference signals for SAP-RPT-1-OSS, weak signals for TabDPT, and no signal for nanoTabPFN.

## Strengths

**The paper studies an important and timely problem.**
As tabular foundation models are increasingly trained on real datasets and deployed behind APIs, dataset provenance and privacy auditing are important practical concerns. The paper addresses a clear gap: how to determine whether a whole tabular dataset was used during TFM pretraining.

**The blind baseline is a strength.**
The authors run the same inference procedure on a TFM that should not have seen the relevant datasets. This helps separate true membership leakage from distributional differences between member and non-member datasets.

**The experiments are careful and nuanced.**
The method works strongly for SAP-RPT-1-OSS, weakly for TabDPT, and not at all for nanoTabPFN. This variation makes the paper more credible than a universal success claim.

**The ablations are informative.**
The paper studies pretraining corpus size, model size, and real versus synthetic pretraining data. The finding that real and real-derived datasets are much more detectable than diverse prior-synthetic datasets is interesting and useful.

## Areas for Improvement

**The assumptions on reference datasets could be made more operational.**
The paper includes useful ablations on pretraining data composition and uses blind baselines to control for distributional differences. However, it would be helpful to discuss more concretely how many known reference datasets an auditor needs, how they should be selected, and how performance changes when (D_{known}) is smaller or distributionally mismatched.

**The blind baseline is good, but its implications could be discussed further.**
The authors correctly report excess over the blind baseline as the relevant leakage signal. However, in some settings the blind baseline itself achieves non-trivial AUC, suggesting that member and non-member datasets are partly distinguishable even without true membership leakage. The paper would benefit from more discussion of what drives this separability and how it affects practical audit reliability.

**The practical audit interpretation could be strengthened.**
ROC-AUC and the external-audit p-values are useful, but a real audit ultimately requires making a decision for a single suspect dataset. The paper could provide clearer guidance on decision thresholds, false-positive rates at fixed p-value levels, and how strong the evidence should be before an auditor treats a dataset as likely included.

## Justification of Score

I would rate this paper a **7**. The problem is important, the method is thoughtful, and the experiments include strong controls such as blind baselines and useful ablations. My main reservations are practical: external auditing is strongly demonstrated for one target TFM, and deployment depends on suitable reference datasets and calibrated thresholds. Overall, this is a good workshop paper and I would accept it.

---

### Official Review · Reviewer_w3vb · 2026-05-22
**A timely and novel topic on dataset inference in TFMs**

**Rating:** 8
**Confidence:** 3

**Review:**

Summary: This paper formulates the task Dataset inference in the context of TFMs, to audit data provenance and privacy leakage. Authors construct dataset fingerprints using 19 input manipulations and extracting 14 output signals across 11 statistical reductions.

strengths:

1. The topic is timely and novel, and highly relevant to the community. The method design provides a tool to audit black-box TFMs. The use of a blind baseline isolates the leakage from distributional differences.
2. The results show the effectiveness of the proposed method.

Area for improvement:

1. The work could benefit more from towards understanding the impact of data leakage on the generalization ability of real unseen data distributions. This could lead to an unbiased evaluation of the performance of these TFMs.